

# Morning surge in blood pressure and sympathetic activity in Mongolians and Han Chinese: a multimodality investigation of hypertension and dyssomnia

Guanhua Huang[1], Xiaoming Yang[2] and Jing Huang[1]

[1] Department of Cardiology, The Second Affiliated Hospital of Chongqing Medical University, Chongqing, China
[2] Department of Cardiology, The Second Affiliated Hospital of Baotou Medical College, Baotou, Inner Mongolia, China

## ABSTRACT

**Background.** Hypertension and dyssomnia are increasing significantly in Mongolians, and the related factors of ethnic differences in hypertension and dyssomnia between Mongolians and Han Chinese are unclear. This study examined the relationship of morning surge in blood pressure (MBP) with ethnicity, sleep situation, and sympathetic activity throughout the day.

**Methods.** Of 692 hypertensive patients screened, 202 subjects with dyssomnia were selected. They were then divided into Mongolian ($n = 87$) and Han ($n = 115$) groups. The differences in dyssomnia, 24-h blood pressure, and urinary catecholamine were analyzed in all subjects; they were then further divided according to the degree of dyssomnia (low, moderate, and severe) to determine the differences in blood pressure and catecholamine.

**Results.** Mongolians had a lower history of smoking, daytime dysfunction, nocturnal heart rates, and dopamine levels, but their body mass index, triglyceride, fasting glucose, morning surge in systolic blood pressure (MSBP), nocturnal systolic blood pressure (NSBP), nocturnal diastolic blood pressure, daytime systolic blood pressure, daytime heart rates, and dopamine level (D-DA) were higher than those of Han Chinese. With the aggravation of dyssomnia, MSBP, NSBP, D-NE, daytime epinephrine, and D-DA of Mongolians and Han Chinese increased gradually, but the rate of increase was faster in the latter ($p < 0.05$). D-DA was entered into the MSBP regression model of Mongolians (intercept, 157 mmHg), whereas D-DA and D-NE were entered into the MSBP regression model of Han Chinese (intercept, 142 mmHg).

**Conclusion.** Worsened dyssomnia induces higher MSBP and augments sympathetic excitability in Mongolians and Han Chinese. Mongolians with hypertension and dyssomnia had higher MSBP baseline and D-DA but lower N-DA. With an increase in D-DA, MSBP in Han and Mongolian patients increased gradually.

Corresponding author
Jing Huang,
huangjing_003@sina.com,
huangjingcqmu@126.com

## INTRODUCTION

Evidence from previous studies suggests that dyssomnia can increase the incidence of high blood pressure, even if it is a single factor that can lead to high blood pressure in a given environment for quite a time (*Joyner, Charkoudian & Wallin, 2010*). However, discomfort caused by elevated blood pressure at night can also lead to sleep disorder (*Walters & Rye, 2009*; *Wing et al., 2010*; *Zou et al., 2009*). Hypertension and dyssomnia seem to be reciprocal causations under certain conditions. At present, patients with these conditions are increasing. However, despite the evidence, physicians often ignore sleep disorders and their possible relationship with hypertension.

Elevated morning surge in blood pressure (MBP) is undoubtedly a major sign of these two diseases, which could increase damage to organs affected by cardiovascular and cerebrovascular diseases and increase morbidity and mortality (*Hoshide et al., 2016*; *Lyhne et al., 2015*; *Sheppard et al., 2015*). Meanwhile, some studies have shown that elevated MBP is related to sympathetic overactivity and the destruction of the body's normal blood pressure rhythm, and it is associated with alterations in sympathetic excitability during sleep (*Johnson et al., 2016*; *Okada et al., 2013*).

In China, different ethnic groups have different levels of blood pressure, and the prevalence of hypertension is also different. Mongolians are the ninth largest minority in China, and their incidence of hypertension is higher than that of Han Chinese (*Okada et al., 2013*). At present, hypertension and dyssomnia in Mongolians are increasing significantly (*Li et al., 2016a*), and the related factors of ethnic differences in hypertension and dyssomnia between these two populations are unclear.

Thus, we conducted a prospective cross-sectional study to confirm the hypothesis that Mongolian patients with hypertension and dyssomnia have more obvious increase in sympathetic excitability, higher MBP, and closer correlation between sympathetic activity and MBP, compared with Han patients.

## MATERIAL AND METHODS

### Study participants

Participants were recruited from the Second Affiliated Hospital of Baotou Medical College in the city of Baotou, Inner Mongolia, by posting study flyers in cardiovascular clinics and wards. Study samples who met eligibility criteria were screened from a total of 692 participants (Fig. 1). The program was approved by the Ethics Committee of the hospital (Ethical Application Ref: 012-2016), and all subjects provided written informed consent.

Hypertensive Mongolian and Han patients $\geq$ 18 years or older with systolic blood pressure (SBP) $\geq$ 140 mmHg and/or diastolic blood pressure (DBP) $\geq$ 90 mmHg and blood pressure < 180/110 mmHg measured three times or more on different days and have not taken any medicines during the week were included.

Patients with secondary hypertension, stroke, coronary heart disease, diabetes mellitus, severe arrhythmia, heart failure, valvular disease, liver and kidney dysfunction, mental disorders, tumor, respiratory diseases, hyperthyroidism, connective tissue disease, or infection; those who had surgery in the past three months; those who were pregnant or

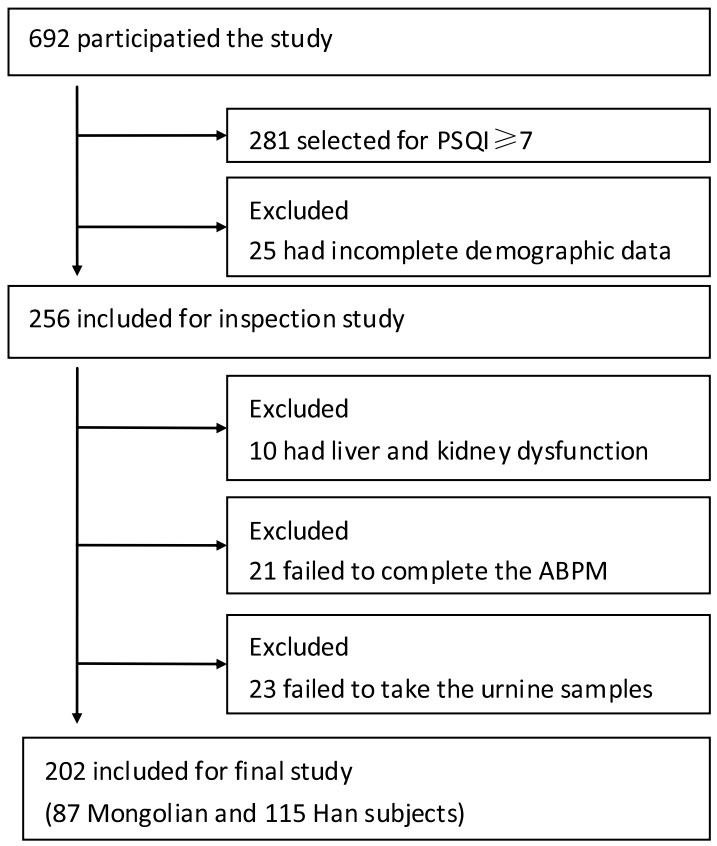

**Figure 1** **Study flow chart of subjects.** ABPM, ambulatory blood pressure monitoring.

breastfeeding; those who cannot independently complete the questionnaire independently; and those who were unwilling to participate in the clinical study were excluded.

To avoid interference factors as much as possible, patients taking any medicines within a week, including antihypertensive drugs and drugs for the treatment of dyssomnia, were excluded. Patients with SBP ≥ 180 mmHg or DBP ≥ 110 mmHg were also excluded to ensure their safety during examination.

## Study methods

All subjects answered the Pittsburgh sleep quality index (PSQI) questionnaire, with PSQI ≥ 7 points being enrolled in this study. Selected subjects supplied their demographic characteristics, and underwent blood biochemistry analysis, 24-h urinary catecholamine detection, and 24-h ambulatory blood pressure monitoring (ABPM). Blood biochemical analysis, 24-h urine catecholamine detection, and ABPM were performed on the same day. Fasting blood was drawn from the subjects before 07:00. The subjects proceeded to the laboratory to get containers and preservatives for collection of 24-h urine catecholamines and then underwent ABPM. The next day, at the appointed time, subjects returned for another ABPM and submission of urine specimens.

## PSQI questionnaire

All subjects were investigated by PSQI, which is a standardized subjective questionnaire used to retrospectively rate sleep quality during the past four weeks (*Buysse et al., 1989*). This instrument is composed of seven components: sleep time, sleep quality, sleep efficiency, sleep latency, sodium amytal use, sleep disturbance and daytime dysfunction. Each category can score up to 3 points, and the instrument's total score can range from 0 to 21 points: 0–6, no sleep disorder; 7–11, mild sleep disorder (PSQI-I); 12–16, moderate sleep disorder (PSQI-II); 17–21, severe sleep disorder (PSQI-III). PSQI $\geq$ 7 is the diagnostic standard of dyssomnia (*Buysse et al., 1989*).

## Demographic feature collection

The demographic characteristics of all selected subjects were collected (name, age, sex, ethnicity, residence, nature of work, exercise, height, weight, and history of smoking). The body mass index (BMI) was computed as weight/height$^2$ (kg/m$^2$).

## Blood index detection

Morning fasting blood was used for the biochemical examination of fasting glucose (FG), triglyceride (TG), total cholesterol (CHO), low-density lipoprotein cholesterol (LDL-C), high-density lipoprotein cholesterol (HDL-C), aspartate alanine transaminase (AST), alanine transaminase (ALT), and creatinine (Cr).

## ABPM

A noninvasive blood pressure monitor (HEM-9000AI; Omron, Kyoto, Japan) that can record the corresponding blood pressure and heart rate was used for the ABPM of subjects every 30 min during daytime (06:00–22:00) and every 60 min during nighttime (22:00–06:00). Daytime systolic and diastolic, nocturnal systolic and diastolic, morning surge in systolic and diastolic blood pressure (DSBP, DDBP, NSBP, NDBP, MSBP, and MDBP, respectively) and the daytime and nocturnal mean heart rates (D-HR and N-HR, respectively) were recorded. Effective blood pressure and heart rate data should be greater than 85% of the set number; otherwise, they were retested.

## Measures of 24-h urine catecholamine

Subjects collected 24-h urine samples during two periods (day and night) of the ABPM session. Each urine sample container had a preservative to prevent deterioration of urine and was stored in portable coolers. Urinary norepinephrine, epinephrine, and dopamine were tested by high-pressure liquid chromatography with electrochemical detection.

## Statistical analysis

Data were expressed as mean $\pm$ standard deviation (SD) for continuous variables and as number (percentage) for categorical variables. To assess ethnic differences, Mann–Whitney U-test was used for continuous non-normally distributed variables and $\chi^2$ test was used for categorical variables. Kruskal-Wallis H-test was used to compare blood pressure and catecholamine in the PSQI groups. Spearman's correlations were used to assess the relationships between MSBP and catecholamine. Multiple linear regression analysis was used to determine the regression equation of MSBP and catecholamine in the Mongolian

and Han groups. The SPSS 22.0 system (IBM, Armonk, NY, USA) was used for the statistical analysis, and significance was set at $p < 0.05$.

## RESULTS

### Patient demographic and clinical characteristics

A total of 692 hypertensive patients were enrolled in this study, of whom 281 (40.61%) had dyssomnia. Because of incomplete registration of subjects, liver and kidney dysfunctions, nonstandardized ABPM data, and failed urine catecholamine specimens, 69 patients were excluded, and finally, 202 complete data of hypertensive patients with dyssomnia were included in the study (Han, $n = 115$, 56.93%; Mongolians, $n = 87$, 43.07%) (Fig. 1). The Mongolian group had higher BMI, FG, TG, DSBP, MSBP, NSBP, NDBP, D-HR, and daytime dopamine level (D-DA) ($p < 0.05$), but their smoking history, DDBP, N-HR, and N-DA were lower than those of the Han group ($p < 0.05$) (Table 1). The changes in SBP, DBP, and HR in Mongolian and Han patients during 24 h were shown in Fig. 2.

### Demographic and basic characteristics of different PSQI groups

There was no statistical difference in the three PSQI degrees between the Mongolian and Han groups in the following aspects: sex, age, residence, nature of work, and movement condition. However, an incidence rate of hypertension of approximately 50% complicated by dyssomnia was found in patients younger than 50 years in these two groups (Table 2).

Sleep time, sleep quality, sleep efficiency, sleep latency, sodium amytal use, sleep disturbance, and PSQI scores had no significant difference between the Mongolian and Han groups, but daytime dysfunction in the Mongolian group was much lower than that of the Han group ($p < 0.01$) (Table 3).

### Comparison of ABPM between the Mongolian and Han patients in different PSQI groups

In the Han group, with gradual aggravation of dyssomnia, DSBP, NSBP, NDBP, and MSBP also increased gradually, and there were statistically significant differences among the three PSQI groups ($p < 0.01$). Meanwhile, DDBP and MDBP also increased gradually, but there was no statistical difference (Table 4).

In the Mongolian group, with gradual aggravation of sleep disorder, NSBP and MSBP also increased gradually, and there were statistically significant differences among the three PSQI groups. Although DSBP, DDBP, NDBP, and MDBP also increased gradually, there was no statistical difference (Table 5).

In the PSQI-I group, DSBP, NSBP, NDBP, and MSBP were significantly different between the Mongolian and Han groups. In the PSQI-II group, DSBP, NSBP, and MSBP had statistically significant differences. In the PSQI-III group, blood pressure had no statistical difference (Table 6).

**Table 1  Demographic and clinical characteristics of study subjects.**

|  | Mongolian group ($n = 87$) | Han group ($n = 115$) | $Z$ ($\chi^2$) | $P$ |
|---|---|---|---|---|
| **Demographics** | | | | |
| Age | $52.01 \pm 10.62$ | $53.23 \pm 9.99$ | $-1.105$ | 0.269 |
| Sex | | | | |
| Male | 40 (34.78%) | 34 (39.08%) | 0.394 | 0.530 |
| female | 47 (65.12%) | 81 (60.92%) | | |
| BMI (kg/m$^2$) | $26.64 \pm 2.59$ | $23.82 \pm 1.99$ | $-7.233$ | 0.000 |
| Smoker (%) | 19 (21.84%) | 31 (27.00%) | 15.380 | 0.000 |
| **Biochemical indexes (mmol/L)** | | | | |
| FG | $5.18 \pm 0.64$ | $4.99 \pm 0.85$ | $-2.328$ | 0.019 |
| CHO | $4.82 \pm 0.93$ | $4.68 \pm 1.06$ | $-1.358$ | 0.174 |
| TG | $2.18 \pm 0.89$ | $1.69 \pm 0.70$ | $-4.002$ | 0.273 |
| HDL-C | $1.49 \pm 0.62$ | $1.43 \pm 0.55$ | $-0.672$ | 0.501 |
| LDL-C PSQI (%) | $2.36 \pm 0.97$ | $2.56 \pm 1.02$ | $-1.097$ | 0.651 |
| PSQI-I | 42 (48.28%) | 57 (49.46%) | 0.033 | 0.856 |
| PSQI-II | 26 (29.89%) | 29 (25.22%) | 0.545 | 0.461 |
| PSQI-II | 19 (21.83%) | 29 (25.22%) | 0.312 | 0.576 |
| **BP (mmHg)** | | | | |
| DSBP | $136 \pm 4$ | $131 \pm 8$ | $-5.592$ | 0.000 |
| DDBP | $86 \pm 8$ | $89 \pm 9$ | $-2.761$ | 0.006 |
| NSBP | $127 \pm 7$ | $119 \pm 12$ | $-5.204$ | 0.000 |
| NDBP | $82 \pm 8$ | $80 \pm 10$ | $-2.588$ | 0.000 |
| MSBP | $163 \pm 7$ | $155 \pm 12$ | $-5.542$ | 0.000 |
| MDBP | $101 \pm 6$ | $102 \pm 8$ | $-1.353$ | 0.176 |
| **HR (beats/min)** | | | | |
| D-HR | $75 \pm 7$ | $73 \pm 5$ | $-0.254$ | 0.024 |
| N-HR | $62 \pm 6$ | $65 \pm 5$ | $-3.031$ | 0.002 |
| **Catecholamine ($\mu$mol L$^{-1}$)** | | | | |
| D-NE | $0.32 \pm 0.12$ | $0.31 \pm 0.16$ | $-1.336$ | 0.181 |
| D-E | $0.22 \pm 0.07$ | $0.21 \pm 0.09$ | $-0.768$ | 0.443 |
| D-DA | $1.14 \pm 0.35$ | $1.01 \pm 0.42$ | $-2.726$ | 0.006 |
| N-NE | $0.23 \pm 0.09$ | $0.22 \pm 0.11$ | $-1.490$ | 0.136 |
| N-E | $0.18 \pm 0.06$ | $0.18 \pm 0.07$ | $-0.140$ | 0.889 |
| N-DA | $0.98 \pm 0.22$ | $1.07 \pm 0.25$ | $-2.499$ | 0.012 |

**Notes.**

For continuous variables, data were mean $\pm$ standard deviation (SD); For categorical variables, data were relative frequencies (percentages).

BMI, body mass index; FG, fasting blood glucose; CHO, total cholesterol; HDL-C, high density lipoprotein; LDL-C, low density lipoprotein; TG, triglyceride; PSQI-I, mild sleep disorder; PSQI-II, moderate sleep disorder; PSQI-III, severe sleep disorder; DSBP, daytime systolic blood pressure; DDBP, daytime diastolic blood pressure; NSBP, nocturnal systolic blood pressure; NDBP, nocturnal diastolic blood pressure; MSBP, morning surge systolic blood pressure; MDBP, morning surge diastolic blood pressure; D-NE, daytime norepinephrine; D-E, daytime epinephrine; D-DA, daytime dopamine; N-NE, nocturnal norepinephrine; N-E, nocturnal epinephrine; N-DA, nocturnal dopamine; HR, heart rates; D-HR, daytime heart rates; N-HR, nocturnal heart rates.

Huang et al. (2017), *PeerJ*, DOI 10.7717/peerj.3758

Peer↲

**Table 2  Demographic characteristics of Mongolian and Han patients among PSQI groups.**

| Group | Mongolian | | | | | | | | Han | | | | | | | |
|---|---|---|---|---|---|---|---|---|---|---|---|---|---|---|---|---|
| | Number | Male | Age < 50 | Age 50–60 | Age > 60 | Living in town | Mainly brainwork | Movement < 1 h/d | Number | Male | Age < 50 | Age 50–60 | Age > 60 | Living in town | Mainly brainwork | Movement < 1 h/d |
| PSQI-I | 42 | 18 (42.86) | 24 (57.14) | 11 (26.19) | 7 (16.67) | 31 (73.81) | 26 (61.90) | 30 (71.43) | 57 | 20 (35.09) | 28 (49.12) | 13 (22.81) | 16 (28.07) | 45 (78.95) | 33 (57.89) | 37 (64.91) |
| PSQI-I | 26 | 10 (38.46) | 13 (50.00) | 7 (26.92) | 6 (23.08) | 20 (76.92) | 19 (73.08) | 20 (76.92) | 29 | 12 (41.38) | 14 (48.28) | 9 (31.03) | 6 (20.69) | 22 (75.86) | 20 (68.97) | 17 (58.62) |
| PSQI-II | 19 | 6 (31.58) | 9 (47.37) | 5 (26.32) | 5 (26.32) | 13 (68.42) | 12 (63.16) | 13 (68.42) | 29 | 8 (27.59) | 15 (51.72) | 8 (27.59) | 6 (20.69) | 21 (72.41) | 21 (72.41) | 19 (65.52) |
| $\chi^2$ | | 0.705 | 0.624 | 0.005 | 0.871 | 1.988 | 1.191 | 0.677 | | 1.221 | 0.078 | 0.720 | 0.850 | 3.664 | 2.127 | 0.398 |
| P | | 0.703 | 0.732 | 0.998 | 0.647 | 0.370 | 0.551 | 0.713 | | 0.543 | 0.962 | 0.698 | 0.654 | 0160 | 0.345 | 0.820 |

**Notes.**

For categorical variables, data were number of the participates and brackets are percentages.

PSQI-I,  mild sleep disorder;  PSQI-II,  moderate sleep disorder;  PSQI-III,  severe sleep disorder; 1 h/d, 1 h per day.

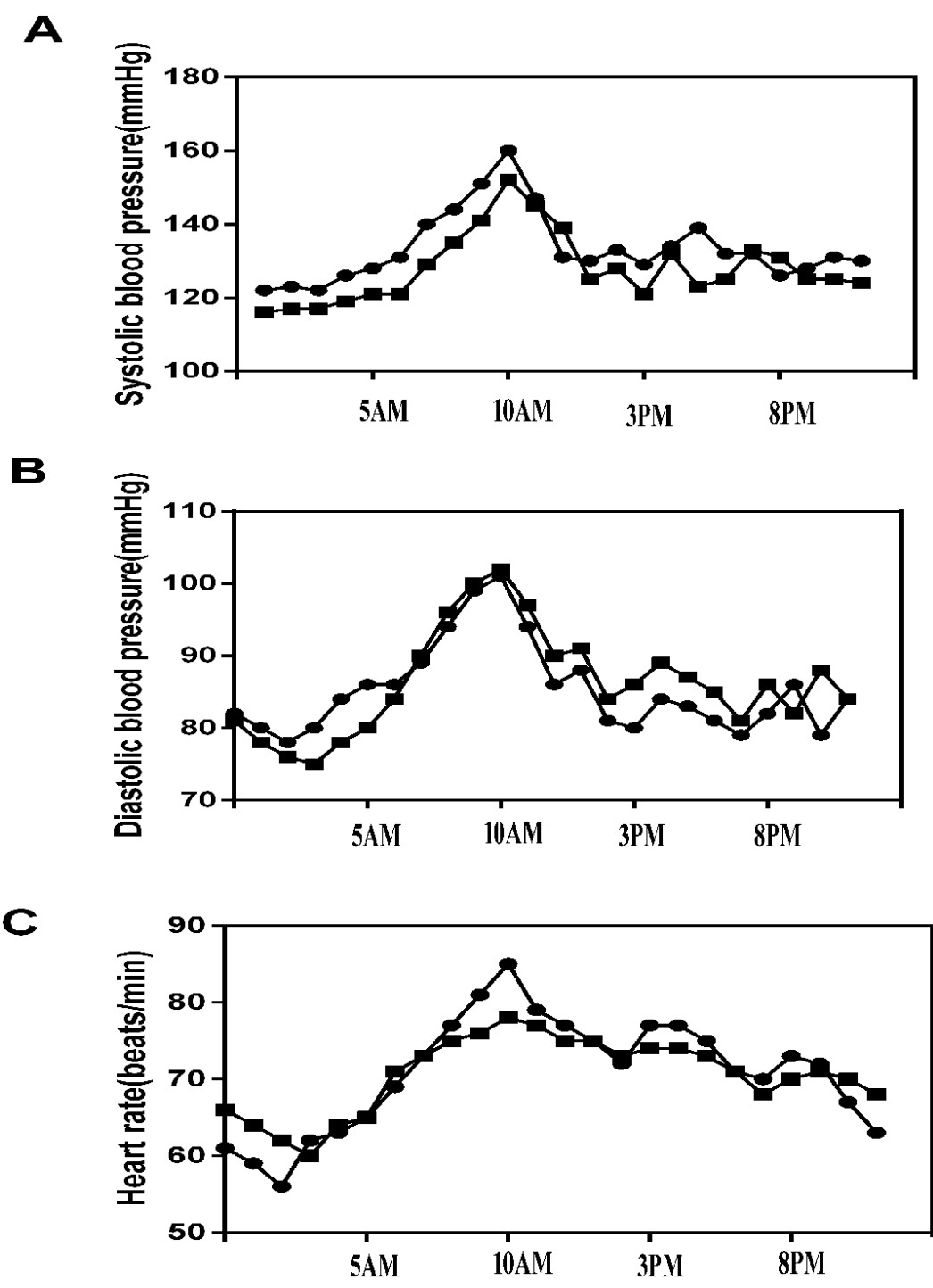

**Figure 2   24-h blood pressure and heart rate between the Mongolian and Han groups.** (A) 24-h systolic blood pressure (SBP) between the Mongolian and Han groups; (B) 24-h diastolic blood pressure (DBP) between the Mongolian and Han groups; (C) 24-h heart rate (HR) between the Mongolian group (circles) and the Han group (squares).

**Table 3   PSQI scores between Mongolian and Han groups.**

|  | Mongolian group ($n = 87$) | Han group ($n = 115$) | $Z$ | $P$ |
|---|---|---|---|---|
| Sleep quality | $2.08 \pm 0.88$ | $2.03 \pm 0.86$ | $-0.484$ | 0.629 |
| Sleep time | $2.18 \pm 0.88$ | $2.23 \pm 0.88$ | $-0.515$ | 0.606 |
| Sleep latency | $2.18 \pm 0.87$ | $2.23 \pm 0.88$ | $-0.515$ | 0.606 |
| Sleep efficiency | $2.16 \pm 0.76$ | $2.09 \pm 0.77$ | $-0.731$ | 0.465 |
| Sleep disturbance | $1.14 \pm 0.98$ | $1.06 \pm 0.97$ | $-0.558$ | 0.577 |
| Sodium amytal use | $1.00 \pm 1.02$ | $0.94 \pm 1.04$ | $-0.507$ | 0.612 |
| Daytime disfunction | $1.82 \pm 0.74$ | $2.10 \pm 0.77$ | $-2.565$ | 0.010 |
| PSQI scores | $12.67 \pm 4.40$ | $12.81 \pm 4.42$ | $-0.262$ | 0.793 |

**Notes.**
For continuous variables, data were mean $\pm$ standard deviation (SD).

## Comparison of 24-h urinary catecholamine between the Mongolian and Han patients in different PSQI groups

In the Han group, with gradual aggravation of dyssomnia, D-NE, D-E, D-DA, N-NE, and N-DA were also increasing and had statistically significant differences among the three PSQI groups (Table 4) ($p < 0.01$).

In the Mongolian group, with gradual aggravation of sleep disorder, N-NE, D-E, and D-DA increasing gradually, but not seen a consistent increasing in D-DA, N-E and N-DA. Meanwhile, with the increase in PSQI degree, the increase in catecholamine was faster in the Han group than in the Mongolian group (Tables 4 and 5).

In the PSQI-I group, D-NE, N-NE, and D-DA were significantly different between the Mongolian and Han groups. In the PSQI-II group, N-DA had statistical difference. In the PSQI-III group, D-NE and N-DA had statistical difference (Table 6).

## Relationship between MSBP and urinary catecholamine

Spearman's correlation of MSBP and D-NE, D-E, D-DA, N-NE, N-E, and N-DA showed that MSBP and D-DA were correlated ($r = 0.254, p < 0.05$, Fig. 3) in the Mongolian group. In the Han group, MSBP was correlated with D-NE, D-E, D-DA, N-NE, and N-DA ($r = 0.396$, $r = 0.285$, $r = 0.403$, $r = 0.309$, and $r = 0.294$, respectively, $p < 0.05$, Fig. 3). Multivariate linear regression analysis was used to rule out the impact of other confounders (age, sex, BMI, smoking, blood sugar, and blood lipid). The regression equation was $y = 157.117 + 5.554x$ ($y$, MSBP; $x$, D-DA) in the Mongolian group, which indicated that when 1 µmol/L D-DA was added, MSBP increased by an average of 5.554 mmHg. In the Han group, the regression equation was $y = 141.730 + 8.108x_1 + 16.984x_2$ ($y$, MSBP; $x_1$, D-DA; $x_2$, D-NE), which suggested that when 1 µmol/L D-DA was added, MSBP increased by an average of 8.108 mmHg, and when 1 µmol/L D-NE was added, MSBP increased by 16.984 mmHg.

## DISCUSSION

Mongolians mainly live in the Inner Mongolia Autonomous Region in China, where Han Chinese and Mongolians account for about 96% of the total population, and these two

Huang et al. (2017), *PeerJ*, DOI 10.7717/peerj.3758

**Table 4  The comparison of ABPM and urinary catecholamine in Han nationality among different PSQI groups.**

| Group | Number | DSBP (mmHg) | DDBP (mmHg) | NSBP (mmHg) | NDBP (mmHg) | MSBP (mmHg) | MDBP (mmHg) | D-NE $\mu$mol L$^{-1}$ | D-E $\mu$mol L$^{-1}$ | D-DA $\mu$mol L$^{-1}$ | N-NE $\mu$mol L$^{-1}$ | N-E $\mu$mol L$^{-1}$ | N-DA $\mu$mol L$^{-1}$ |
|---|---|---|---|---|---|---|---|---|---|---|---|---|---|
| PSQI-I | 57 | 128 ± 9 | 87 ± 9 | 112 ± 11 | 78 ± 6 | 150 ± 11 | 101 ± 7 | 0.20 ± 0.08 | 0.18 ± 0.08 | 0.75 ± 0.23 | 0.15 ± 0.05 | 0.17 ± 0.07 | 0.96 ± 0.24 |
| PSQI-I | 29 | 132 ± 5 | 90 ± 6 | 122 ± 8 | 80 ± 15 | 154 ± 7 | 104 ± 7 | 0.35 ± 0.09 | 0.23 ± 0.07 | 1.05 ± 0.20 | 0.27 ± 0.11 | 0.18 ± 0.06 | 1.08 ± 0.19 |
| PSQI-III | 29 | 136 ± 4 | 91 ± 7 | 130 ± 5 | 83 ± 9 | 167 ± 8 | 104 ± 9 | 0.48 ± 0.17 | 0.27 ± 0.10 | 1.50 ± 0.41 | 0.29 ± 0.12 | 0.21 ± 0.07 | 1.27 ± 0.21 |
| $\chi^2$ | | 31.942 | 4.491 | 53.453 | 11.796 | 46,462 | 4.482 | 61.855 | 17.662 | 58.111 | 36.346 | 4.788 | 26.628 |
| P | | 0.000 | 0.106 | 0.000 | 0.003 | 0.000 | 0.106 | 0.000 | 0.000 | 0.000 | 0.000 | 0.091 | 0.000 |

**Notes.**

For continuous variables, data were mean ± standard deviation (SD).

PSQI-I, mild sleep disorder; PSQI-II, moderate sleep disorder; PSQI-II, severe sleep disorder; DSBP, daytime systolic blood pressure; DDBP, daytime diastolic blood pressure; NSBP, nocturnal systolic blood pressure; NDBP, nocturnal diastolic blood pressure; MSBP, morning surge systolic blood pressure; MDBP, morning surge diastolic blood pressure; D-NE, daytime norepinephrine; D-E, daytime epinephrine; D-DA, daytime dopamine; N-NE, nocturnal norepinephrine; N-E, nocturnal epinephrine; N-DA, nocturnal dopamine.

Huang et al. (2017), *PeerJ*, DOI 10.7717/peerj.3758

**Table 5** The comparison of ABPM and urinary catecholamine in Mongolian among different PSQI groups.

| Group | Number | DSBP (mmHg) | DDBP (mmHg) | NSBP (mmHg) | NDBP (mmHg) | MSBP (mmHg) | MDBP (mmHg) | D-NE $\mu$mol L$^{-1}$ | D-E $\mu$mol L$^{-1}$ | D-DA $\mu$mol L$^{-1}$ | N-NE $\mu$mol L$^{-1}$ | N-E $\mu$mol L$^{-1}$ | N-DA $\mu$mol L$^{-1}$ |
|---|---|---|---|---|---|---|---|---|---|---|---|---|---|
| PSQI-I | 42 | 135 ± 3 | 85 ± 7 | 125 ± 5 | 81 ± 9 | 162 ± 8 | 100 ± 7 | 0.27 ± 0.09 | 0.20 ± 0.05 | 1.03 ± 0.33 | 0.22 ± 0.08 | 0.17 ± 0.06 | 0.99 ± 0.25 |
| PSQI-II | 26 | 136 ± 3 | 86 ± 9 | 127 ± 8 | 82 ± 7 | 163 ± 7 | 101 ± 5 | 0.37 ± 0.11 | 0.24 ± 0.08 | 1.07 ± 0.17 | 0.22 ± 0.09 | 0.19 ± 0.06 | 0.93 ± 0.21 |
| PSQI-III | 19 | 136 ± 4 | 88 ± 8 | 131 ± 6 | 83 ± 7 | 168 ± 6 | 102 ± 7 | 0.35 ± 0.14 | 0.24 ± 0.08 | 1.48 ± 0.40 | 0.25 ± 0.12 | 0.18 ± 0.06 | 1.00 ± 0.17 |
| $\chi^2$ | | 0.170 | 1.914 | 14.471 | 0.231 | 7.622 | 0.232 | 13.606 | 9.022 | 17.715 | 1.214 | 1.686 | 0.765 |
| P | | 0.918 | 0.384 | 0.001 | 0.891 | 0.022 | 0.890 | 0.001 | 0.011 | 0.000 | 0.545 | 0.431 | 0.682 |

**Notes.**

For continuous variables, data were mean ± standard deviation (SD).

PSQI-I, mild sleep disorder; PSQI-II, moderate sleep disorder; PSQI-II, severe sleep disorder; DSBP, daytime systolic blood pressure; DDBP, daytime diastolic blood pressure; NSBP, nocturnal systolic blood pressure; NDBP, nocturnal diastolic blood pressure; MSBP, morning surge systolic blood pressure; MDBP, morning surge diastolic blood pressure; D-NE, daytime norepinephrine; D-E, daytime epinephrine; D-DA, daytime dopamine; N-NE, nocturnal norepinephrine; N-E, nocturnal epinephrine; N-DA, nocturnal dopamine.

Huang et al. (2017), *PeerJ*, DOI 10.7717/peerj.3758

**Table 6 The comparison of ABPM and urinary catecholamine between Mongolian and Han nationality in different PSQI groups.**

| Group | Ethnicity | Number | DSBP (mmHg) | DDBP (mmHg) | NSBP (mmHg) | NDBP (mmHg) | MSBP (mmHg) | MDBP (mmHg) | D-NE $\mu$mol L$^{-1}$ | D-E $\mu$mol L$^{-1}$ | D-DA $\mu$mol L$^{-1}$ | N-NE $\mu$mol L$^{-1}$ | N-E $\mu$mol L$^{-1}$ | N-DA $\mu$mol L$^{-1}$ |
|---|---|---|---|---|---|---|---|---|---|---|---|---|---|---|
| PSQI-I Mongolian | Mongolian | 42 | 135 ± 3 | 85 ± 7 | 125 ± 5 | 81 ± 9 | 162 ± 8 | 100 ± 7 | 0.27 ± 0.09 | 0.20 ± 0.05 | 1.03 ± 0.33 | 0.22 ± 0.08 | 0.17 ± 0.06 | 0.99 ± 0.25 |
| | Han | 57 | 128 ± 9 | 87 ± 9 | 112 ± 11 | 78 ± 6 | 150 ± 11 | 101 ± 7 | 0.20 ± 0.08 | 0.18 ± 0.08 | 0.75 ± 0.23 | 0.15 ± 0.05 | 0.17 ± 0.07 | 0.96 ± 0.24 |
| Z | | | −6.188 | −1.652 | −6.855 | −2.697 | −5.947 | −0.121 | −3.866 | −1.359 | −4.272 | −4.560 | −0.238 | −0.397 |
| P | | | 0.000 | 0.099 | 0.000 | 0.007 | 0.000 | 0.904 | 0.000 | 0.174 | 0.000 | 0.000 | 0.812 | 0.691 |
| PSQI-II | Mongolian | 26 | 136 ± 3 | 86 ± 9 | 127 ± 8 | 82 ± 7 | 163 ± 7 | 101 ± 5 | 0.37 ± 0.11 | 0.24 ± 0.08 | 1.07 ± 0.17 | 0.22 ± 0.09 | 0.19 ± 0.06 | 0.93 ± 0.21 |
| | Han | 29 | 132 ± 5 | 90 ± 6 | 122 ± 8 | 80 ± 15 | 154 ± 7 | 104 ± 7 | 0.35 ± 0.09 | 0.23 ± 0.07 | 1.05 ± 0.20 | 0.27 ± 0.11 | 0.18 ± 0.06 | 1.08 ± 0.19 |
| Z | | | −3.317 | −1.749 | −2.559 | −1.793 | −4.079 | −1.793 | −0.734 | −0.658 | 0.261 | −1.662 | −0.938 | −2.429 |
| P | | | 0.002 | 0.080 | 0.011 | 0.073 | 0.000 | 0.073 | 0.463 | 0.510 | 0.794 | 0.096 | 0.348 | 0.015 |
| PSQI-III | Mongolian | 19 | 136 ± 4 | 88 ± 8 | 131 ± 6 | 83 ± 7 | 168 ± 6 | 102 ± 7 | 0.35 ± 0.14 | 0.24 ± 0.08 | 1.48 ± 0.40 | 0.25 ± 0.12 | 0.18 ± 0.06 | 1.00 ± 0.17 |
| | Han | 29 | 136 ± 4 | 91 ± 7 | 130 ± 5 | 83 ± 9 | 167 ± 8 | 104 ± 9 | 0.48 ± 0.17 | 0.27 ± 0.10 | 1.50 ± 0.41 | 0.29 ± 0.12 | 0.21 ± 0.07 | 1.27 ± 0.21 |
| Z | | | −0.616 | −1.123 | −0.603 | −0.433 | −0.445 | −0.782 | −2.489 | −0.749 | −0.084 | −0.971 | −1.478 | −3.819 |
| P | | | 0.538 | 0.261 | 0.546 | 0.665 | 0.656 | 0.434 | 0.013 | 0.454 | 0.933 | 0.332 | 0.118 | 0.000 |

**Notes.**

For continuous variables, data were mean ± standard deviation (SD).

PSQI-I, mild sleep disorder; PSQI-II, moderate sleep disorder; PSQI-III, severe sleep disorder; DSBP, daytime systolic blood pressure; DDBP, daytime diastolic blood pressure; NSBP, nocturnal systolic blood pressure; NDBP, nocturnal diastolic blood pressure; MSBP, morning surge systolic blood pressure; MDBP, morning surge diastolic blood pressure; D-NE, daytime norepinephrine; D-E, daytime epinephrine; D-DA, daytime dopamine; N-NE, nocturnal norepinephrine; N-E, nocturnal epinephrine; N-DA, nocturnal dopamine.

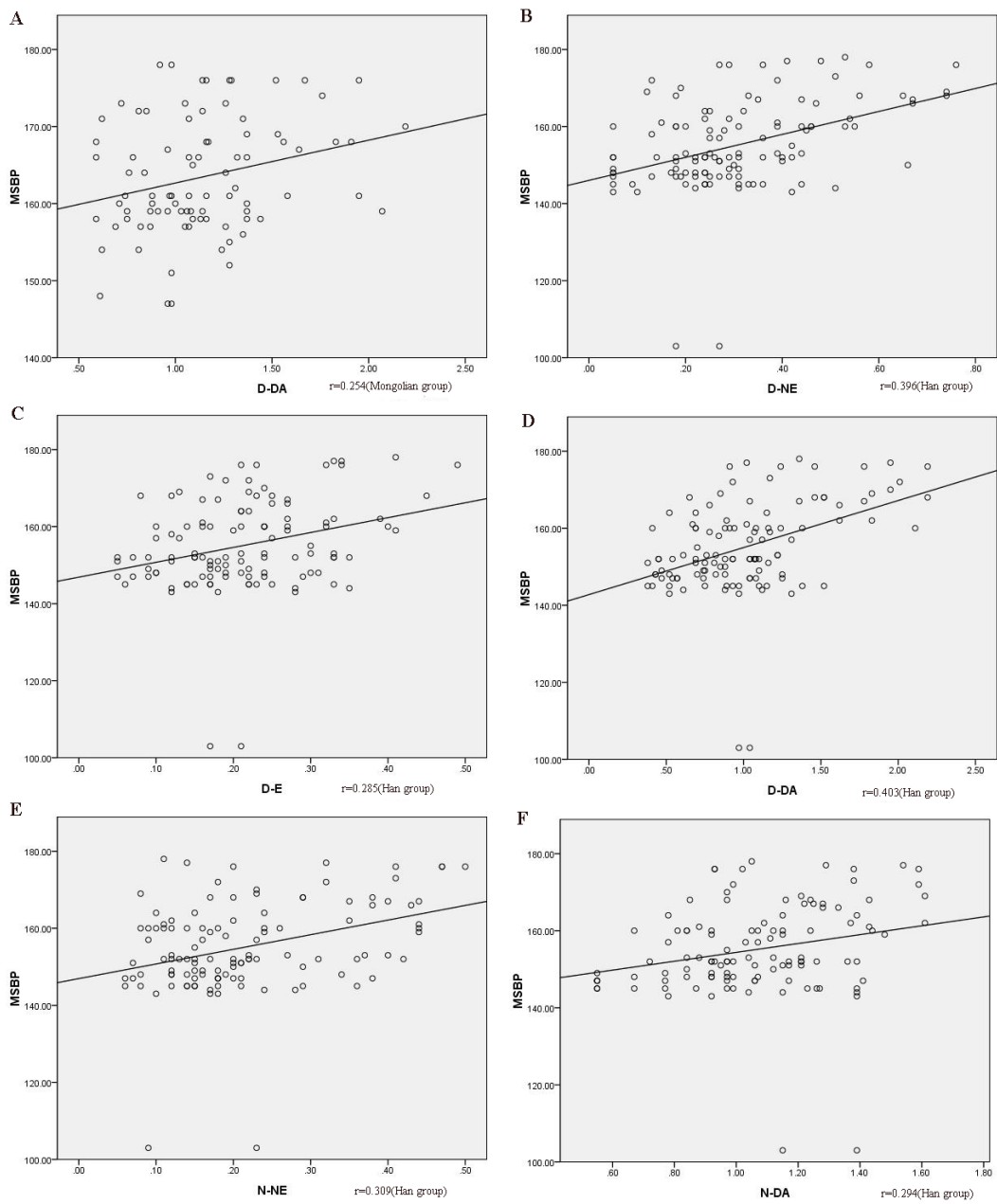

**Figure 3** **The correlation between MSBP and catecholamines.** (A) the correlation between MSBP and D-DA in the Mongolian group; (B) the correlation between MSBP and D-NE in the Han group; (C) the correlation between MSBP and D-E in the Han group; (D) the correlation between MSBP and D-DA in the Han group; (E) the correlation between MSBP and N-NE in the Han group; (F) the correlation between MSBP and N-DA in the Han group. MSBP, morning surge systolic pressure; MSBP, morning surge systolic blood pressure; D-DA, daytime dopamine; D-NE, daytime norepinephrine; D-E, daytime epinephrine; N-NE, nocturnal norepinephrine; N-DA, nocturnal dopamine.

ethnic groups have different customs and lifestyle (*Okada et al., 2013*). There is marked ethnic difference in the prevalence of hypertension. A cross-sectional study in, 2014 showed that the prevalence rate of hypertension in the Inner Mongolia Autonomous Region was higher in Mongolians (31.46%) than in Han Chinese (27.47%) (*Li et al., 2016a*). Previous reports suggested that obesity, dietary history, habit, and genetics were all hypertension-related risk factors in the Mongolian population (*Carretero & Oparil, 2000*; *Vaidya & Forman, 2010*). Hypertension is considered as the major cause of disability and death worldwide, especially MSBP, which is associated with coronary heart disease, stroke, and renal dysfunction (*Hu et al., 2012*; *Kearney et al., 2005*). The relationship between BP and the risk of cardio-cerebrovascular disease is continuous—the risk of developing myocardial infarction, heart failure, and stroke increases with higher BP (*Li et al., 2008*). Mongolians in China were found to have higher prevalence of hypertension and other cardiovascular risk factors than the Han Chinese (*Li et al., 2016b*). In addition, the prevalence rates of hypertensive complications such as CVD and stroke are significantly higher in the same geographical area than in the Han Chinese (*Zhang et al., 2007*).

## Characteristics of Mongolian and Han patients

As can be seen in our study, Mongolians had higher BMI, FG, and TG levels than Han Chinese, but a lower history of smoking, which can probably be attributed to their carnivorous diet and low preference for smoking. In our opinion, elevated BMI, FG, and TG levels depend on dietary history to a great extent and on some habits, such as high-salt diets and alcohol consumption, which are both closely associated with dietary history. Smoking, a commonly accepted hypertension-related factor (*Carretero & Oparil, 2000*), was lower in the Mongolian group than in the Han group in the present study, which is contrary to the high incidence of hypertension in Mongolians. There were no differences in residence, nature of work, or movement between the two groups, possibly because the recruited addresses were located in a central urban area.

In addition, this study showed that even with the same degree of dyssomnia at night, the Mongolian group had lower daytime dysfunction and physical discomfort. We also found that regardless of the level of sleep disorder, the highest incidence of hypertension complicated with dyssomnia was found in patients younger than 50 years in these two groups, which accounted for half of the subjects. Consistent with previous reports, middle-aged, rather than elderly, Chinese with sleep disorder were more likely to be hypertensive than healthy people (*Guo et al., 2016*). Similar disparities according to age have also been found in other countries (*Dean et al., 2012*; *Lombardi & Parati, 2000*).

## Sleep and sympathetic nerve activity

Sleep has been recognized as an essential element of cardiovascular control, because sleep disorders are interrelated with increased risk of hypertension, cardiac disease, and stroke (*Cappuccio et al., 2011*; *Dean et al., 2012*; *Lombardi & Parati, 2000*). Sleep-related hypertension is often masked as a non-dipping status at an early stage. Thereafter, an increase in sympathetic activity results in daytime hypertension and elevated MSBP. Normal nocturnal sleep is characterized by a marked decrease in sympathetic activity and

blood pressure, which affects the sympathetic rhythm of blood pressure during the following daytime period (*Sayk et al., 2007*). Deeper sleep has been shown to be accompanied by further suppression of the sympathetic function. Lack of an enough deeper sleep period can lead to sympathetic nervous system dysfunction (*Yang et al., 2002*). Some studies showed that dyssomnia was related to changes in sympathovagal activity, as evidenced by increased catecholamine release and decreased heart rate (*Lusardi et al., 1999*; *Spiegel, Leproult & Van Cauter, 1999*), which were similar to the findings of our study.

In our study, with the aggravation of sleep disorder, whether in the Mongolian or Han group, DSBP, DDBP, NSBP, NDBP, MSBP, and MDBP gradually increased as well, which was in agreement with previous studies (*Haas et al., 2005*; *Meisinger et al., 2007*). Furthermore, the blood pressure of the Han group was more sensitive to the degree of dyssomnia and blood pressure increased more significantly. Meanwhile, with the aggravation of dyssomnia, N-NE, D-E, and D-DA increased as well. The catecholamine level of the Han group increased obviously during both daytime and nighttime, whereas the increasing catecholamine of the Mongolian group did not have consistent with the degree of PSQI. Besides, in the PSQI-I group, the catecholamine of the Mongolian group was higher than that of the Han group at both daytime and nighttime; in the PSQI-II group, the catecholamine of the Mongolian group was higher than that of the Han group during daytime, but lower at nighttime; in the PSQI-III group, regardless of time, the catecholamine of the Mongolian group was lower. It seems that the rate of catecholamine secretion was reversed between the two groups with aggravation of sleep disorder. This trend was similar, but not identical, to that of blood pressure in our study. Different degrees of sleep disorder affect the change in MSBP, either in Mongolian or Han patients, and the higher the degree of sleep disorder, the higher the MSBP. Increasing evidence suggests that sleep and sleep-related blood pressure control have a significant contribution to the development of hypertension (*Carrington et al., 2005*; *Carrington & Trinder, 2008*; *Lombardi & Parati, 2000*).

## Blood pressure and sympathetic nervous activity

The sympathetic nervous system is thought to play a crucial role in the regulation of blood pressure during daytime and nighttime. During sleep, the baroreceptor reflex is lower, which reduces the sympathetic activity of the heart and blood vessels, eventually leading to a drop in blood pressure (*DiBona, 2002*). However, sympathetic overactivity in sleep disorder causes elevated blood pressure and diurnal blood pressure control dysfunction, which lead to damage of target organs.

Previous research indicated that the sympathetic nervous system had a significant influence on MSBP (*Kario et al., 2004*; *Hashimoto et al., 2003*). Otherwise, individuals with exaggerated MSBP have higher levels of urinary catecholamine excretion (*Hashimoto et al., 2003*). In our study, there was a positive correlation between D-DA and MSBP in Mongolian and Han patients. Regression analysis showed that D-DA was independently correlated with MSBP. The two sets of the regression intercepts were 157 and 142 mmHg in Mongolians and Han Chinese, respectively, which showed that the change in D-DA affected MSBP and the starting point of MSBP in Mongolian patients was

higher by about 15 mmHg than that of Han patients. With increasing D-NE, MSBP was also increasing in Han patients, but this was not seen in Mongolian patients. Studies showed that catecholamine may be pivotal in the racial variation in blood pressure, which suggested that African-Americans had better receptor sensitivity than European-Americans (*Mills et al., 1995*; *Rutledge, 1991*). Some observational research have shown that in real life, the level of variations in catecholamine during the day is related to the degree of decrease in blood pressure with sleep (*Arita et al., 1996*).

### Study limitations

Our study has limitations, such as its small sample size. Additionally, we did not use polysomnography to assess sleep architecture. Furthermore, the Mongolian patients in our study may have changed their habits and customs to a certain extent when they lived in the same area as the Han Chinese, and high blood pressure therapy may have been underestimated due to inclusion criteria of the study.

## CONCLUSIONS

Worsened dyssomnia induces higher MSBP and augments sympathetic excitability in Mongolians and Han Chinese. However, Mongolian patients have higher baseline of MSBP but lower daytime dysfunction. They also have higher D-DA but lower N-DA. With the increase in D-DA, the MSBP in Mongolian patients also gradually increased, but their rate of increase is lower than that in Han patients. These ethnic characteristics might provide information when deciding on clinical medication. Further studies on the mechanism of underlying genetic factors and pathological routines in the two ethnicities are needed.

## ACKNOWLEDGEMENTS

We appreciate the staff of the Institute of Hypertension and Biochemical Laboratory in the Second Affiliated Hospital of Baotou Medical College.

### Funding

The study was funded by the the Inner Mongolia Autonomous Region Natural Science Foundation, Project No. 2016MS08112. There was no additional external funding received for this study. The funders had no role in study design, data collection and analysis, decision to publish, or preparation of the manuscript.

### Grant Disclosures

The following grant information was disclosed by the authors:
Inner Mongolia Autonomous Region Natural Science Foundation: 2016MS08112.

### Competing Interests

The authors declare there are no competing interests.

## Author Contributions

- Guanhua Huang conceived and designed the experiments, performed the experiments, analyzed the data, contributed reagents/materials/analysis tools, wrote the paper, prepared figures and/or tables.
- Xiaoming Yang performed the experiments, analyzed the data, contributed reagents/materials/analysis tools, prepared figures and/or tables, reviewed drafts of the paper.
- Jing Huang conceived and designed the experiments, wrote the paper, reviewed drafts of the paper.

## Human Ethics

The following information was supplied relating to ethical approvals (i.e., approving body and any reference numbers):

The Second Affiliated Hospital of Baotou Medical College granted ethical approval to carry out the study within its facilities (Ethical Application Ref: 12, 2016).

## Clinical Trial Registration

The following information was supplied regarding Clinical Trial registration:

http://www.chictr.org.cn/listbycreater.aspx

ChiCTR-ROC-16010182.

## Data Availability

Raw data can be found in the Supplemental Information.

## Supplemental Information

Supplemental information for this article can be found online at http://dx.doi.org/10.7717/peerj.3758#supplemental-information.

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
