# Peer review of "Morning surge in blood pressure and sympathetic activity in Mongolians and Han Chinese: a multimodality investigation of hypertension and dyssomnia"

_PeerJ, doi:10.7717/peerj.3758_

## Round 0.1 · original submission · Major Revisions

Both the reviewers and I are concerned about the correlation and regression analysis between urinary catecholamines and MSSBP. The detail in the analysis should be provided and I suggest to present these data as Figures. Figures 2 and 3 can be converted into table(s).
The conclusion was not made based on the results, please carefully the conclusion and indicate its clinical significance.

Reviewer 1 ·

Basic reporting

Major comments:
1. Basically, essential evidences show the different prevalence of hypertension related complications between these two groups should be included in this paper, such as coronary artery disease, stroke and renal dysfunction.
2. The results are displayed without logistic description. In table 1, only D-DA and N-DA are different between the two groups, which could not generalized to the difference of catecholamine or sympathetic activity. In table 4, increased blood pressure with increased PSQI score in both groups, however, not consistent with increased catecholamine or sympathetic activity. R values in the regression analysis are too low even though the p-values show differences.
3. The language of the manuscript should be carefully polished.

Minor points:
1. In the first paragraph of discussion, important evidence “citation 8”, which shows different hypertension prevalence in these two groups, is not cited properly. So as other numbered citations in the paper.
2. Abbreviations in the paper are organized improperly.
3. Amount of useless data in tables which have never described in the main content.

Experimental design

In this study, after figuring out higher prevalence of hypertension in the Mongolians than the Han Chinese, the authors raised a good question about morning surge systolic blood pressure and autonomic nerve, especially sympathetic activity in these two groups. A large number of subjects (202 effective in 692) are involved in this study and this study provides some preliminary observations and several marked differences between these two groups. The topic is meaningful and may indicate a direction to figure out the difference between these two groups.

Validity of the findings

Based on the data shown, the authors could not get the conclusions in the paper.

Additional comments

No comment

Reviewer 2 ·

Basic reporting

no comment.

Experimental design

no comment.

Validity of the findings

no comment.

Additional comments

In this article Huang et al conducted a prospective cross-sectional study to confirm the hypothesis that Mongolian patients with hypertension and dyssomnia have more obvious increase in sympathetic excitability, higher MSBP, and closer correlation between sympathetic activity and MSBP, compared with Han patients. Their findings indicated that in hypertension and dyssomnia, Han Chinese were more sensitive to changes in catecholamine, but Mongolians had higher MSSBP baseline. It’s a novel and interesting finding, whereas it needs to be improved.


Major
Comment 1. Elevated MSBP and dyssomnia are associated with sympathetic overactivity, hence sympathetic nervous system activity is a key index in this study. The authors measured sympathetic nervous system activity only through analysing the catecholamine in 24h urinary. As we known, increases the heart rate is related to the sympathetic nervous system enhancement, thus it is necessary to add heart rate, heart rate variability and other indicators that are related to SNS. Moreover, it is best to visually reflect 24h blood pressure, heart rate changes and make real-time comparisons when drawing. Detailed mapping methods should be consulted related paper which were published.
(Morning surge in blood pressure is associated with reactivity of the sympathetic nervous system.Am J Hypertens. 2014 Jun;27(6):783-92.)

Comment 2. Table 2 the authors used the χ2 test to analyze the Demographic and basic characteristics of PSQI. In my opinion, the 2X3 crosstabs analysis may be more suitable.

Comment 3. Multiple linear regression analysis was used to determine the regression equation of the MSSBP and catecholamine in the Mongolian and Han groups. More details should be provided. Did the authors ruled out the impact from other confounder (smoking? )?

Minor:
1. In the structure ABSTRACT, Discussion or Conclusion?

2. The title INRRODUCTION should be “INTRODUCTION”

3. In line 53 “At present, hypertension and dyssomnia in Mongolians are increasing significantly......” a references about hypertension and dyssomnia in Mongolians are increasing significantly should be added.

4. In line 94.......dyssomnia(Buysse et al. 1989), the form of reference annotation is different from the other parts of the text.

---

## Round 0.2 · Minor Revisions

Reviewer 1 is concerned about the inconsistency between results and conclusions. Please carefully make any conclusions and discussion of results based on your findings in this study. Also, please respond to the other concerns of Reviewer 1

The minor revisions raised by reviewer 2 should also be corrected. Please also keep in mind that authors are responsible for the English language.

Reviewer 1 ·

Basic reporting

Compared with the former version, the paper is failed to be rewritten in professional English.

Experimental design

The authors failed to get three important aims in the paper:
1. Worsen dyssomnia induces higher MSBP in each group. Even though it is described in the discussion, the tables and figures did not show any comparison within groups. For example, MSBP increased with PSQI scores in each group, p value should be provided, as well as other indicators.
2. Worsen dyssomnia augments sympathetic excitability within and between groups.
3. Comparison between the Han and Mongolia groups, while absence of proper epidemiological references, endpoint measurements for complications of evaluated MSBP or hypertension are essential in this paper. The following is the contradiction between the discussion and the conclusion.
[According to the conclusion, Han people are more sensitive to worsen dyssomnia than the Mongolian people which leads to rapid evaluation of MSBP. “The Mongolian people in China were found to have higher prevalence of hypertension and other cardiovascular risk factors than the Han community (Li et al., 2016b). (in the first paragraph of discussion)”, however, “sympathetic overactivity in sleep disorder causes elevated blood pressure and diurnal blood pressure control dysfunction, which lead to damage of target organs. Intrinsic brain and vascular mechanisms may be implicated. Furthermore, sympathetic stimulation contributes to high blood pressure by increasing vascular resistance and cardiac function, altering endothelial function (Schiffrin, 2002), inducing cardiac and vascular remodeling (Schiffrin, 2002), and/or changing the renal sodium and water homeostasis (DiBona, 2002). (Discussion - Blood pressure and sympathetic nervous activity)”]

Validity of the findings

no comment

Additional comments

The raw data shows several interesting findings within each group, stronger evidences should be collected before making conclusion.

Reviewer 2 ·

Basic reporting

-

Experimental design

-

Validity of the findings

-

Additional comments

The authors had made the corresponding revisions, and now the manuscript is much improved but I still have some minor comments as below,
1. In Table 1, Female in Han group, 81(60.92), is a percent sign “%” missing here?
2. Legend of table 1, DDBP=daytime diastolic pressure, the full name should be daytime diastolic blood pressure. Please check the other legend and in manuscript.

---

## Round 0.3 · accepted · Accept

All the reviewers' concerns have been addressed.